# Non-adherence to self-care and associated factors among diabetes adult population in Ethiopian: A systemic review with meta-analysis

Teshager Weldegiorgis Abate[1]*, Getenet Dessie[1], Yinager Workineh[2], Haileyesus Gedamu[1‡], Minyichil Birhanu[2], Emiru Ayalew[1‡], Mulat Tirfie[3‡], Aklilu Endalamaw[2]

1 Department of adult health Nursing, School of Health Science, College of Medicine and Health Science, Bahir Dar University, Bahir Dar, Ethiopia, 2 Department of Pediatric and Child Health Nursing, School of Health Sciences, College of Medicine and Health Sciences, Bahir Dar University, Bahir Dar, Ethiopia, 3 Department of nutrition and dietetics, School of Public Health, College of Medicine and Health Science, Bahir Dar University, Bahir Dar, Ethiopia

☉ These authors contributed equally to this work.
‡ These authors also contributed equally to this work
* teshagerhylemarriam@gmail.com

**Data Availability Statement:** All relevant data are within the manuscript and its Supporting information files.

## Abstract

### Background

Self-care practice among people with diabetes is not well-implemented in Ethiopia. So far, in Ethiopia, several observational studies have been done on self-care practice and its determinants in people with diabetes. However, a comprehensive review that would have a lot of strong evidence for designing intervention is lacking. So, this review with a meta-analysis was conducted to bridge this gap.

### Methods

A systematic review of an observational study is conducted following the PRISMA checklist. Three reviewers have been searched and extracted from the World Health Organization's Hinari portal (SCOPUS, African Index Medicus, and African Journals Online databases), PubMed, Google Scholar and EMBASE. Articles' quality was assessed using the Newcastle-Ottawa Scale by two independent reviewers, and only studies with low and moderate risk were included in the final analysis. The review presented the pooled proportion of non-adherence to self-care practice in people with diabetes and the odds ratios of risk factors hindering to self-care practice after checking for heterogeneity and publication bias. The review has been registered in PROSPERO with protocol number CRD 42020149478.

### Results

We included 21 primary studies (with 7,134 participants) in this meta-analysis. The pooled proportion of non-adherence to self-care in the diabetes population was 49.91 (95% CI:

**Funding:** No specific funding for this work.

**Competing interests:** The authors have declared that no competing interests exist.

44.73–55.08, $I^2$ = 89.8%). Male (Pooled Odds Ratio (POR): 1.84 95%CI; 1.04–2.64, $I^2$ = 15.0%), having private glucometer (POR: 2.71; 95%CI: 1.46–3.95, $I^2$ = 0.0%), short-term Diabetes Mellitus (DM) duration (POR: 3.69; 95%CI: 1.86–5.52, $I^2$ = 0.0%), DM complication (POR: 2.22; 95%CI: 1.48–2.95, $I^2$ = 0.0%), treatment satisfaction (POR: 1.8; 95% CI: 1.15–2.44, $I^2$ = 0.0%), received diabetes self-management education (POR: 2.71; 95% CI: 1.46–3.95, $I^2$ = 0.0%) and poor self-efficacy (POR: 3.09; 95% CI: 1.70–4.48, $I^2$ = 0.0%) were statistically significant factors of non-adherence to self-care practice.

## Conclusions

The overall pooled proportion of non-adherence to self-care among adult diabetes in Ethiopia was high. Further works would be needed to improve self-care practice in the diabetes population. So, factors that were identified might help to revise the plan set by the country, and further research might be required to health facility fidelity and each domain of self-care practice according to diabetes self-management guideline.

## Background

Diabetes Mellitus (DM) is one of the major public health problems worldwide. DM is one of the four priorities on non-communicable diseases increasing by epidemic proportions [1–5]. The number of people living with diabetes will reach 693 million cases in the world by the end of 2045. Of this, 16 to 41 million cases will be in Africa [6,7]. Now a day, DM is recognized as an important cause of premature death and lifelong disability [8,9]. People with diabetes have an increased risk of developing serious life-threatening health problems, increasing medical care costs, and compromise the quality of life [4,6,7,10].

Diabetes brings a global economic burden that will increase from the United State $1.3 trillion in 2015 to $2.1 trillion in the target scenarios by 2030. This translates to an increase in costs as a share of global gross domestic product [11,12]. Health systems in Sub-Saharan Africa countries are unable to cope with the current burden of DM. The unique challenges to combat DM in these regions are lack of [1] fund for non-communicable disease-specific to the diabetes population [2], availability of enough guidelines specific to the diabetes population [3], availability of adequate medications, and [4] service access differences in the urban and the rural person with diabetes and [5] inequity between the public and the private sector of healthcare. Because of such obstacles, diabetes has a greater impact on morbidity and mortality in sub-Saharan Africa as compared to other regions in the world [8,9,11,13–15].

Individuals with DM develop lifelong self-care skills and are committed to making behavioral changes to prevent or delay complications [16,17]. Self-care is the practice of activities that an individual initiates and performs on his or her own behalf to keep up with a healthy life. It is also a way of life, not just a task [18,19]. Adherence to self-care is the extent to which a person's behavior in which following a recommended diet, foot care practice, self-monitoring of plasma glucose, and/or executing lifestyle changes corresponds with agreed recommendations from a healthcare provider [17,20].

The main reason for non-adhered to self-care practice was health care providers, and diabetes educators did not check perceived client barriers to self-care practice and make recommendations with these in mind. Many health care providers are not discussing self-care practice with diabetic people [21–24]. Through multiple barriers to accepting self-care practice, low

health literacy, socioeconomic burden, and social support, factors influence self-care non-adherence in a person with diabetes. In Ethiopia, diabetes self-management programs, including diabetes education, are not accessible where the care is [25,26].

Realizing the multi-faceted nature of the problem, the health care providers and the patient shall build a trusting relationship, a systematic, multi-pronged, and integrated approach is required for promoting self-care practice adherence. Non-adherence to self-care practice in diabetes has been found to vary from region to region in Ethiopia [27–31]. Even though the pooled proportion (49%) of good diabetes self-care behavior among people with diabetes is documented in Ethiopia [32], the overall non-adherence and common factors that hinder self-care practice are not documented in the country. Thus, this study aimed to assess the pooled proportion of non-adherence to self-care and associated factors among the adult diabetes population in Ethiopia.

## Materials and methods

### Protocol design and registration

A systematic review of an observational study is conducted following the meta-analysis of observational studies in an epidemiology statement. The Preferred Reporting Items for Systematic Reviews and Meta-Analyses Protocol (PRISMA—P) [33,34] and Meta-Analysis of Observational Studies in Epidemiology (MOOSE) guideline [35] were used for the development of this study protocol, the conduct and design, and reporting these results (S 1). To minimize duplication of the same reviews, provide transparency, and reduce reporting bias of the current review, the protocol was registered with the International Registration of Systematic Reviews (PROSPERO) with PROSPERO registration number CRD 42020149478 (S 2).

### Eligibility criteria

The studies (all published and unpublished) were that used observational epidemiological designs (cross-sectional), intended to assessed non-adherence to self-care practice and associated factors among people with diabetes aged 15 and above, and articles that were published in the English language. Studies were conducted in a case series, unclear definition of self-care practice (like unclear measurement of questionnaires in the outcome variables, and did not report specific outcomes for self-care adherence/non-adherence quantitatively) were excluded in the final analysis.

### Data sources and searching strategy

A search strategy has been developed using fundamental concepts in the research question: "self-care adherence", "self-care practice", "therapy adherence", "treatment adherence", "medication intake adherence", "medication compliance", "patient compliance", "diabetes mellitus", "diabetes", "patients", "clients", and "factors", "determinants", "influences", "risk factors", "predictors" and "Ethiopia". For each key concept, appropriate free-text words and Medical Subject Heding (MeSH) were used and combined using boolean operators such as "AND' and "OR." This enabled the retrieval of relevant articles that might have used different synonyms for the same word. Notably, to fit the advanced PubMed database, the search was strategy applied (S 3).

A pretest of the search strategy by three authors was performed in PubMed, and the actual electronic search was done between 25 February and March 5, 2020. Three reviewers implemented the electronic search in the following electronic databases: PubMed, Embase, GOOGLE SCHOLAR, CINAHL, MEDLINE, and Hinari electronic databases for articles published

on 17 April 2012 to 10 September 2019. Hinari is the World Health Organization (WHO) database portal for low and middle income countries and includes Web of Science, SCOPUS, African Index Medicus (AIM), Cumulative Index to Nursing and Allied Health Literature (CINAHL), WHO's Institutional Repository for Information Sharing (IRIS), and African Journals Online databases. In addition, articles were also searched through a review of the grey literature available on institutional repositories (Addis Ababa University). Besides, find articles by reviewing the reference lists of already identified researches.

## Study selection

All the citation identified by our search strategy, which was potentially eligible for inclusion, was exported to EndNote software version X7, Thomson Reuters, New York, NY, and the duplicate were removed. Title and abstracts of the remaining citation were screened by three independent reviewers (TWA, EA, and GD) and ineligible studies were further excluded. The full texts of selected articles were retrieved and read thoroughly to ascertain their suitability before data extraction. Articles that fulfilled the earlier criteria used as sources of data for analysis.

## Data extraction

The abstract and full-text review and data abstraction were done by two independent reviewers (HG and MT) using a standardized data abstraction form, developed according to the sequence of variables required from primary studies on MS-Excel sheet. The disagreement between the two reviewers was resolved by a third independent reviewer (GD or AE). Before analysis, a transformation of the Adjusted odds ratios and proportion.

The New castle Ottawa Scale criteria was selected for quality assessment of selected studies before analysis [36]. Two independent reviewers (HG and MT) critically appraised each article using the NOS. Discrepancies between reviewers resolved by discussion and by including a third reviewer (MB). The average of two independent reviewer's quality scores used to decide whether the articles included or not. Articles with methodological flaws or incomplete reporting of results in the full-text excluded from the analysis. The data extraction formant included primary author, publication year, region, outcome measuring tool, study design, response rate, sample size, and proportion.

## Outcome measurement

The primary outcome of this review the pooled proportion of non-adherence to the self-care practice of people with diabetes in Ethiopia. The proportion measured as the number of adult diabetes with non-adherence to self-care practice in the studies divided by the total number of diabetes people in a study multiplied by 100. For the analysis of the secondary outcomes (factors), we extracted data on factors that were related to non-adherence to self-care practice in the literature. Such as social support, having own glucometer at home, duration of diabetes, diabetes-related complication, diabetes knowledge, received diabetes self-management education, and socio-demographic related factors. In examining factors associated with non-adherence to self-care practice, data used from the primary studies of the Adjusted Odd Ratios (AOR) to find the association between the independent variables and having non-adherence to self-care practice.

## Quality assessment

The risk of bias of included studies is assessed using the 10-item rating scale developed by Hoy et al. for prevalence studies [37]. The assessment tool has a representative sample size, method

of data collection, reliability and validity of study tools, case definition, and prevalence periods of the studies. Researchers categorized each study as having a low risk of bias ("yes" answers to domain questions) or a high risk of bias ("no" answers to domain questions). Each study assigned a score of 1 (Yes) or 0 (No) for each domain, and these domain scores were summed to give an overall study quality score. Scores of 8–10 were considered as having a "low risk of bias", 6–7 a "moderate risk", and 0–5 a "high risk" (S 4). For the least risk of bias classification, discrepancies between the reviewers resolved via consensus.

## Statistical analysis

**Testing for heterogeneity.** Heterogeneity between the result of the primary studies was assessed using Cochran's Q test and quantified with the $I^2$ statistics. A p-value of less than 0.1 was considered to suggest statically significant heterogeneity, considering a category a small number of studies and their heterogeneity in design [38]. Heterogeneity had taken low, moderate, and high categories when the $I^2$ values were below 25%, between 25% and 75%, and above 75%, respectively [39,40]. Thus, the random effect model was used to pool the proportion of non-adherence to self-care practice since the studies were found heterogeneous. The random effect model accounts for heterogeneity among study results beyond the variation associated with chance, unlike the fixed-effect model [41]. To investigate the source of heterogeneity, the random-effects meta-regression was conducted by taking primary study characteristics such as study year, region, and study setting (hospitals). The meta-regression analysis was weighted to account for the residual between-study heterogeneity (i.e., heterogeneity not explained by the covariates in the regression) [42]. Subgroup analyses by region and type of study setup (hospitals) were carried out because of significant heterogeneity between studies (i.e., $I^2$ = 89.8%, p < 0.05).

**Publication bias assessment.** Publication bias was assessed by visual inspection of funnel plots based on the shape of the graph (subjective assessment). The symmetrical graph was interpreted to suggest an absence of publication bias, whereas an asymmetrical one indicates the presence of publication bias. On the other hand, qualitatively (objective ealuation), Egger's weighted regression tests was used to assess publication bias with a p-value less than 0.1 considered as indicative of a statistically significant publication bias [43].

**Sensitivity analysis.** Lastly, a sensitivity analysis was done to estimate whether the pooled effect size was affected by single studies. A leave-one-out sensitivity analysis was performed to confirm whether there were studies that potentially biased the direction of the pooled estimate.

## Result

### Study selection

The database search and desk review yielded a total of 914 articles. Of these, 896 articles were retrieved from PubMed, Google Scholar, EMBASE, and the World Health Organization's Hinari portal (which includes the SCOPUS, African Index Medicus, and African Journals Online databases). The remaining 18 observational studies were found from institutional repositories (Addis Ababa, Gondar, and Bahir Dar Universities). After reviewing the titles and abstracts, we excluded 510 articles due to duplication. In screening, we excluded 382 articles because their outcomes were not in line with the desired eligibility criteria. The full-text of the remaining 22 articles were accessed for eligibility and quality. Additionally, an article was excluded because its outcomes variable was not clearly stated [44] (S 5). The remaining 21 studies were included in the analysis (Fig 1).

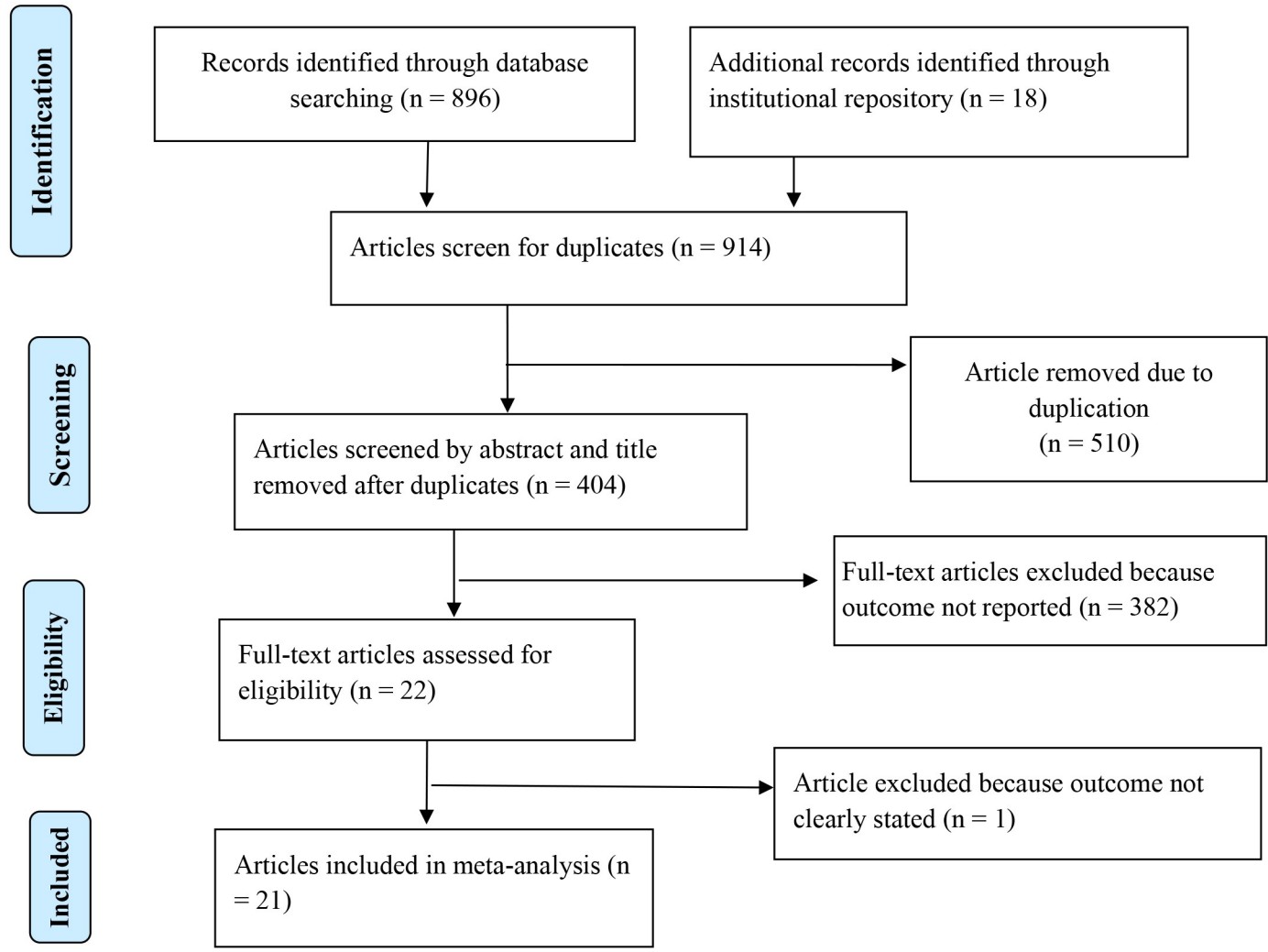

**Fig 1. PRISMA statement presentation for a meta-analysis of a pooled proportion of non-adherence to self-care practice among DM in Ethiopia, 2011–2019.**

## Study characteristics

Overall, a total of 21 observational studies were selected in this systematic review and meta-analysis. This consisted of 7,134 participants (aged 15–86 years). The number of participants in each study ranged from 222 to 595. The mean age ranged from 38 to 55.03 years and the duration of the disease from 1.74 to 12 years. All studies were cross-sectional design to estimate self-care adherence. The most retrieved studies (n = 5) were from Oromia region [45–49], followed by Addis Ababa (n = 4) [50–53], and Amhara region (n = 3) [31,54,55], Eastern Ethiopia (Dire Dawa and Harari) [27,56,57], and Tigray region [29,58,59] were represented by three studies, whereas Southern Nations Nationalities and People's Region (SNNPR) represented by two studies [60,61], and Benishangul Gumez region was represented by one study [62]. Except for two studies [51,52], all the studies reported in peer-reviewed journals are reported in peer-reviewed journals. All of the studies used a summary of diabetes self-care activities to investigate the non-adherence to self-care practice and reported high response rates (> 95%) (Table 1).

**Table 1. Descriptive summary of 21 studies included in the meta-analysis of the proportion of non-adherence to self-care practice in Ethiopia from 2011–2019.**

| Authors name | Study year | Source types | Region | Types of hospital | Age of subjects | RR (%) | SS | Total outcome | Prevalence (%) | Quality score (NOS) |
|---|---|---|---|---|---|---|---|---|---|---|
| Abate TW. et al | 2017 | Journal | Amhara | Referral | ≥18 | 99.5 | 416 | 298 | 71.6 | 8 |
| Addisu Y. et al | 2013 | Journal | SNNPR | Referral | ≥18 | 100 | 310 | 72 | 23.2 | 6 |
| AlemayhuT.et al | 2018 | Journal | Dire Dawa | General | ≥18 | 98.6 | 506 | 223 | 44.1 | 8 |
| Amente T. et al | 2013 | Journal | Oromia | Referral | ≥15 | 98 | 254 | 114 | 45 | 7 |
| AschalewAY. et al | 2017 | Journal | Amhara | Referral | ≥18 | 100 | 403 | 209 | 51.86 | 6 |
| Ayele K. et al | 2011 | Journal | Harai | Referral | ≥18 | 100 | 222 | 135 | 60.8 | 7 |
| Ayele BH.et al | 2017 | Journal | Harar and Dire Dawa | General | ≥18 | 97.8 | 320 | 198 | 61.9 | 8 |
| Berhan T. et al | 2015 | IR | A.A | General | ≥18 | 100 | 595 | 284 | 47.7 | 8 |
| Berhe KK.et al | 2012 | Journal | A.A | Referral | 30–85 | 99.1 | 320 | 142 | 44.4 | 7 |
| Berhe KK. et at | 2013 | Journal | Tigray | Referral | ≥18 | 96.8 | 300 | 147 | 49 | 7 |
| Chali SW. et al | 2018 | Journal | Benishangul Gumuz | General | ≥18 | 96 | 383 | 175 | 45.7 | 8 |
| Dedefo MG.et al | 2016 | Journal | Oromia | Referral | ≥15 | 100 | 252 | 99 | 39.3 | 6 |
| FEYISSA L.et at | 2014 | IR | A.A | Referral | ≥18 | 98.8 | 324 | 157 | 48.5 | 7 |
| Gurmu Y.et al | 2017 | Journal | Oromia | General | ≥18 | 100 | 257 | 117 | 45.5 | 8 |
| Kassahun T.et al | 2014 | Journal | Oromia | Referral | ≥18 | 95 | 309 | 152 | 49.1 | 8 |
| Mamo M. et al | 2011 | Journal | A. A | Referral | ≥18 | 97.8 | 646 | 256 | 39.7 | 8 |
| Mariye T. et al | 2017 | Journal | Tigray | General | ≥18 | 100 | 284 | 178 | 62.7 | 7 |
| Niguse H. et al | 2017 | Journal | Tigray | Referral | ≥18 | 100 | 338 | 252 | 74.5 | 7 |
| Sorato MM.et al | 2015 | Journal | SNNPR | General | ≥15 | 100 | 194 | 114 | 58.8 | 5 |
| Tadesse E. et al | 2014 | Journal | Oromia | Referral | ≥30 | 100 | 116 | 59 | 50.9 | 6 |
| Tiruneh SA.et al | 2018 | Journal | Amhara | General | ≥18 | 95 | 385 | 142 | 36.9 | 8 |

SS: Sample size; RR: Response Rate; IR: Institutional repository; A.A: Addis Ababa, NOS: New-castle Ottaw Scale.

## Quality appraisal

The quality score of the included study ranged from 5 to 8 with a mean score of 7.14 (SD = 0.91). Out of 21 studies, 16 (76.19%) studies received a low risk of bias, and 5 (23.81%) studies received a moderate risk of bias. The authors also find types of bias: 10 studies [63–72] had a high risk of case definition, 8 studies [65,68,71,73–77] had a high risk of sampling and representation [64,68,69,73,74,76,78,79] bias, and 5 studies [66,67,69,71,73] had a high risk of random selection bias (S4).

## Meta-analysis

**Pooled estimates of non-adherence to self-care.** The analysis of twenty-one observational studies was ranked as low and moderate-quality. The pooled proportion of non-adherence to self-care practice in people with diabetes was 49.91% (95%CI: 40.12–55.25, $I^2$ = 89.8). The highest (71.60%) [55] and the lowest (23.20%) [60] non-adherence of self-care practice reported in the Amhara region and in the SNNPR respectively. High heterogeneity was observed among the included studies (Q test P<0.001) and $I^2$ ($I^2$ = 89.8%) (Fig 2). Due to the heterogeneity of included studies, further sub-group analysis was done by using the following study characteristics: regional location and study setting (hospital). The random-effect model was applied for reporting the pooled proportion of non-adherence to the self-care practice of the sub-group analysis.

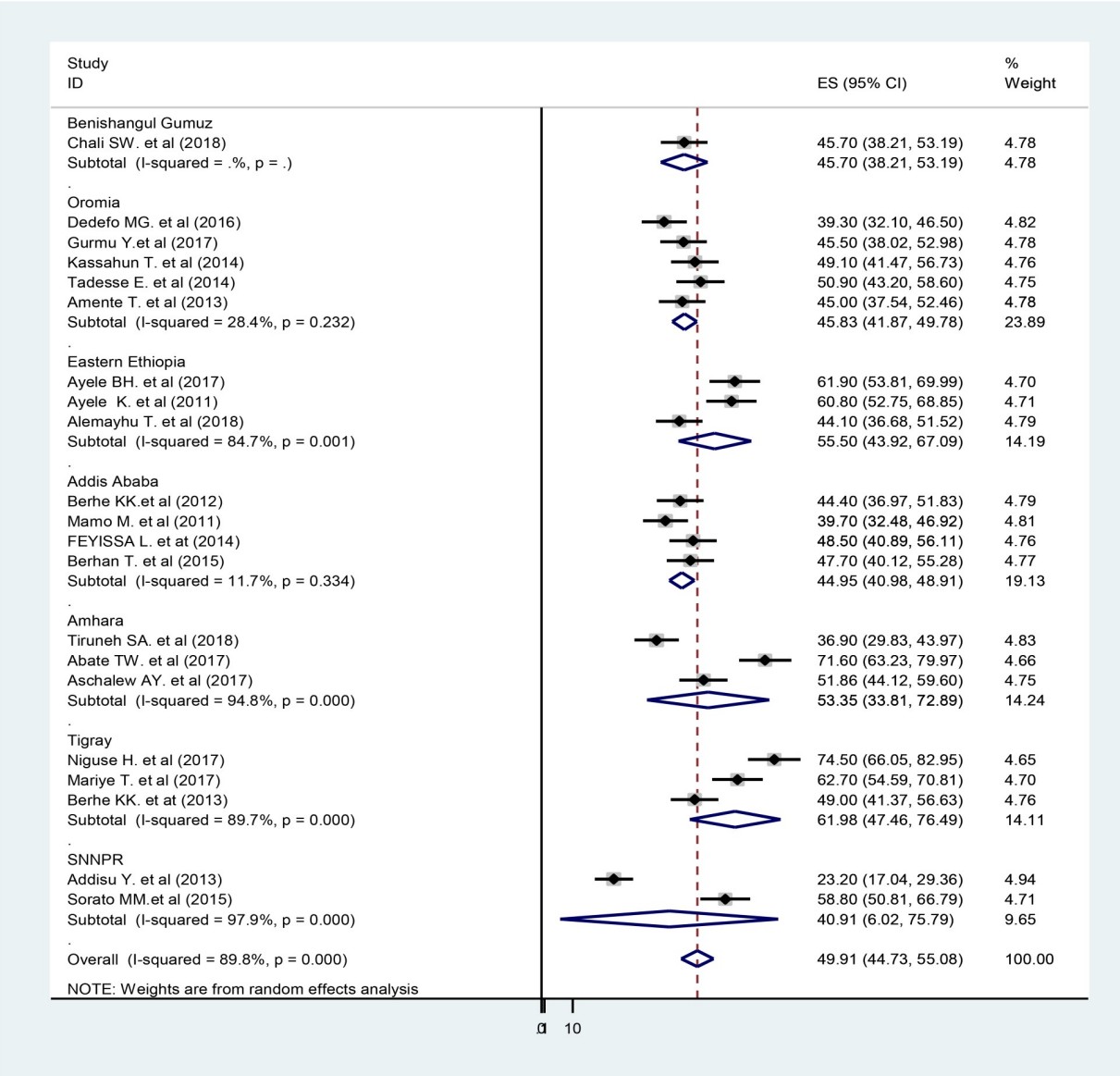

**Fig 2. A subgroup analysis of the forest plot showing the pooled proportion of non-adherence to self-care practice among DM in Ethiopia, 2011–2019.**

**Subgroup analysis.** On subgroup analysis by region, the highest pooled proportion of non-adherence self-care was found in the Tigray region (Pooled Proportion (PP) = 61.98%; 95% CI: 47.46–76.49) and the lowest in the SNNPR (PP = 40.91%; 95% CI: 6.02–75.79). The pooled proportion of non-adherence to elf-care among DM in the Amhara region was 53.35% (95% CI: 33.81, 72.89), and 44.95% (95% CI: 40.98, 48.91) in Addis Ababa. The pooled proportion of non-adherence to self-care practice in a referral hospital is 49.5% (42.20, 57.18) (Table 2).

**Publication bias.** Both funnels plots of precision asymmetry and the Egger's test of the intercept showed that there is no publication bias in the primary studies. Visual examination of the funnel plot showed a symmetric distribution of studies. Additionally, Egger's test of the

**Table 2. Sub-group analysis of non-adherence to self-care practice based on hospital and region in Ethiopia from 2011 to 2019.**

| Variables | Characteristics | Estimated proportion of non-adherence to self-care practice | $I^2$ p value |
|---|---|---|---|
| Region | Benishangul Gumuz | Single study | Single study |
| | Oromia region | 45.83 (41.81, 49.78) | 28.4%, p = 0.232 |
| | Eastern Ethiopia* | 55.50 (43.92, 67.09) | 84.7%, p < 0.001 |
| | Addis Ababa | 44.95 (40.98, 48.91) | 11.7%, p = 0.334 |
| | Amhara | 53.35 (33.81, 72.89) | 98.8%, p <0.001 |
| | Tigray | 61.98 (47.48, 76.49) | 89.7%, p <0.001 |
| | SNNPR | 40.91 (6.02, 75.79) | 97.9%, p <0.001 |
| Hospitals | General | 50.28 (43.74, 55.81) | 82.9%, p <0.001 |
| | Referral | 49.69 (42.20, 57.18) | 92.20%, p <0.001 |

SNNPR = Southern Nations Nationalities and Peoples' Region,

* = Harai and Dire Dawa.

intercept was 0.0786 (95% CI: -0.18, 0.34) p > 0.05 (0.54), as judged by Egger's test, there was no evidence of publication bias being presented at 5% significance level (Fig 3).

**Meta-regression and sensitivity analysis.** The sub-group analysis showed that heterogeneity across the studies was widespread. To name the source of heterogeneity, we conducted a meta-regression and sensitivity analysis. During the meta-regression analysis, we conducted using the following study covariance: publication years, sample size, and region. However, the

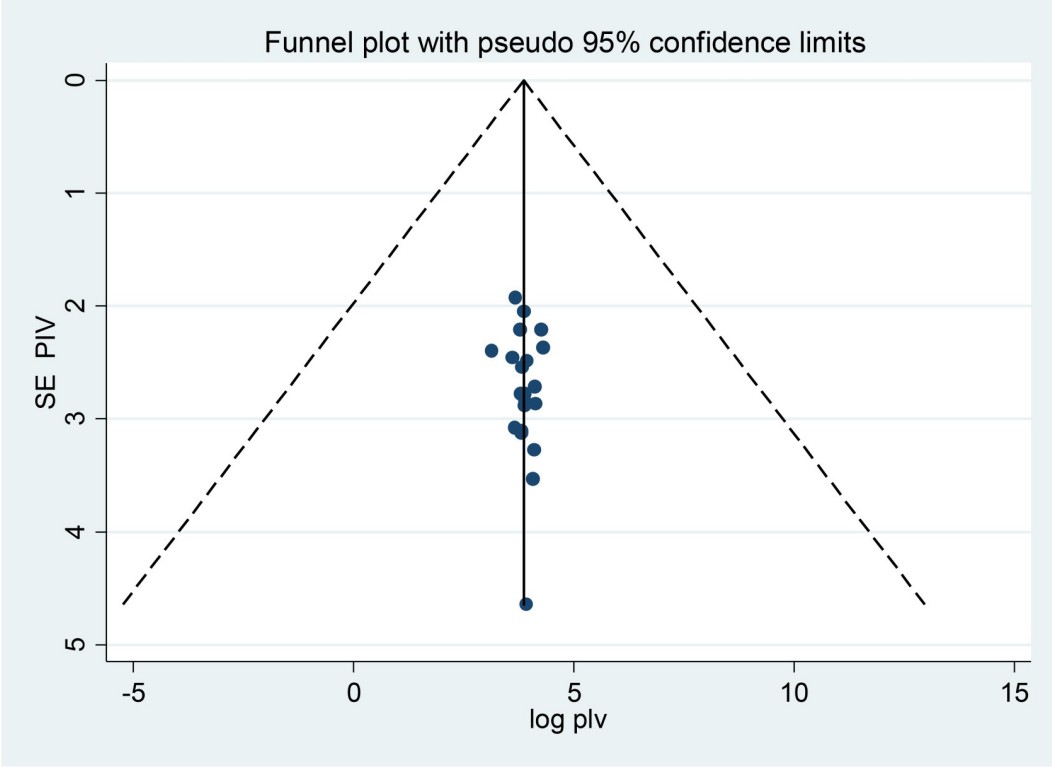

**Fig 3. Meta funnels presentations of the proportion of non-adherence to self-care practice among diabetes in Ethiopia, 2011–2019, whereby SE PIV (standard error of proportion) plotted on the Y-axis and log PIV (logarithm of proportion) on the X-axis.**

**Table 3. Meta-regression output to explore heterogeneity of the pooled proportion of non-adherence to diabetes self-care practice among the diabetes population in Ethiopia from 2011 to 2019.**

| Variables | Coefficients | P-value | 95% CI |
|---|---|---|---|
| Study Year | 0.67 | 0.983 | -63.62, 64.95 |
| Sample size | 0.01 | 0.987 | -1.16, 1.15 |
| Region | | | |
| Benishangul Gumuze | | | Single study |
| Addis Ababa | -0.65 | 0.961 | -28.52, 27.22 |
| Amhara | 9.71 | 0.57 | -20.97, 36.58 |
| Eastern Ethiopia (Harai and Dire Dawa) | 9.79 | 0.479 | -19.04, 38.61 |
| Oromia | 0.19 | 0.989 | -27.22, 27.59 |
| SNNPR | -5.14 | 0.725 | -35.76, 25.51 |
| Tigray | 16.45 | 0.241 | -12.37, 45.27 |

results showed that none of these variables were a statistically significant source of heterogeneity (Table 3). We also performed a sensitivity analysis to find the influence of each study on the overall effect size. No single primary study affected the overall pooled proportion of non-adherence to self-care practice among people with diabetes in Ethiopia (Fig 4).

**Determinants of non-adherence to self-care practice.** Extracted adjusted odds ratios from primary studies were organized into four themes (socio-demographic, clinical, get access to glucometere and DM education, and psychosocial) and pooled to identify the predominant

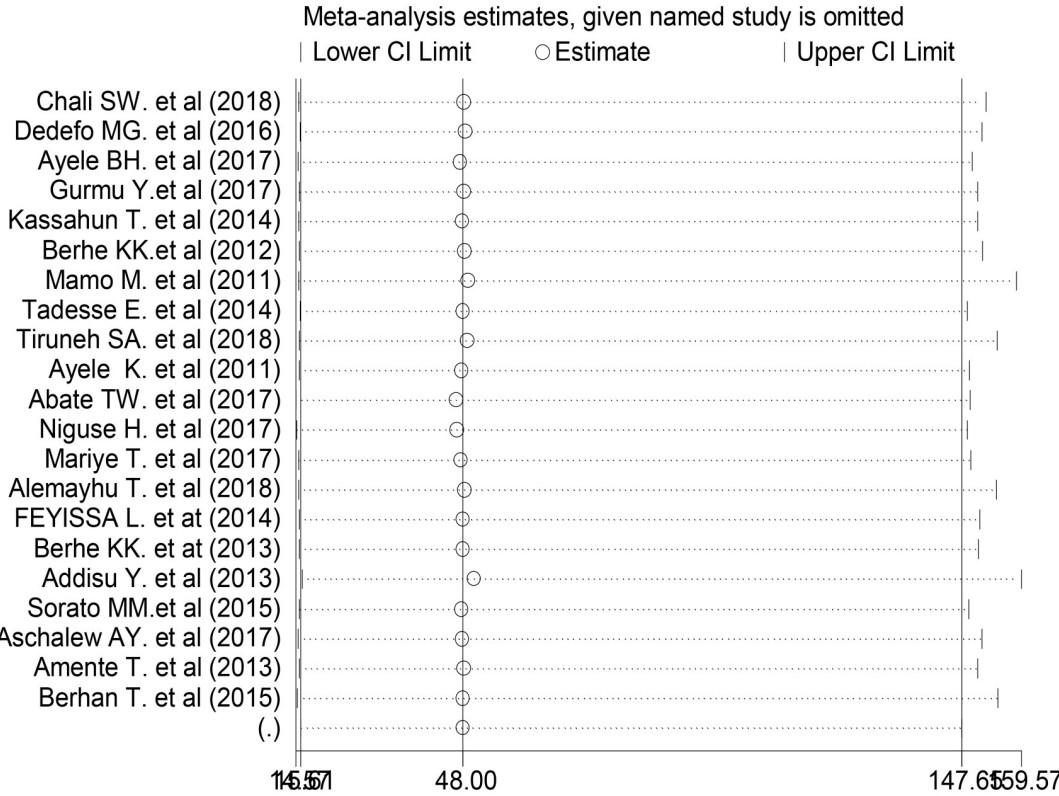

**Fig 4. One-leave-out sensitivity analysis for studies conducted on proportion of non-adherence to self-care practice among people with DM in Ethiopia, 2011–2019.**

**Table 4. The relationship between non-adherence to diabetes self-care practice and the pooled effect of factors eligible for meta-analysis among diabetes population in Ethiopia from 2011–2019.**

| Factors | Non-Adherence to Self-Care Practice | | |
|---|---|---|---|
| | AOR (95% CI) | $I^2$ (%) | P-value (n) |
| Socio-demographic factors | | | |
| Age | 1.49 (-0.94, 3.92) | 51.2 | 0.129 |
| Sex | 1.84 (1.04, 2.64) | 15.0 | 0.803* |
| Residency | 0.48 (-0.26, 1.21) | 0.0 | 0.499 |
| Educational status | 2.89 (1.52, 4.25) | 72.3 | <0.001* |
| Income | 0.85 (0.15, 1.54) | 69.5 | <0.011 |
| Accessibility | | | |
| Access to DSME | 2.76 (1.97, 3.55) | 0.0 | <0.001* |
| Access to glucometer at home | 2.71 (1.46, 3.95) | 0.0 | 0.926* |
| Knowledge toward DM | 1.48 (0.79, 2.17) | 77.3 | <0.001 |
| Clinical related factors | | | |
| DM duration | 3.69 (1.86, 5.52) | 0.0 | 0.631* |
| DM complication | 2.22 (1.48, 2.95) | 0.0 | 0.891* |
| Satisfaction for medical care | 1.8 (1.55, 2.44) | 0.0 | 0.604* |
| Treatment modality | 1.19 (-0.19, 2.57) | 90.4 | <0.001 |
| Psychosocial related factors | | | |
| Socials support | 1.31 (0.72, 1.89) | 86.5 | <0.001 |
| Self-efficacy | 3.09 (1.70, 4.48) | 0.0 | 0.845 |

*Significanty associated; DSME: Diabetes Self-Management Education.

risk factors for non-adherence to self-care practice among diagnosed diabetes population in Ethiopia.

**Socio-demographics factors.** In the primary studies, a significant association was observed between socio-demographic variables and non-adherence to self-care. Such as, participants who had a family history of DM (AOR: 0.5; 95% CI: 0.30,0.90) [47], being widowed (AOR: 4.05; 95% CI: 1.41, 11.5) [55], self-employed (AOR: 5.94; 95%CI: 1.97, 17.94) [29], being single (AOR: 5.39; 95% CI: 1.75, 16.75) [75], and being divorced (AOR: 0.2; 95% CI: 0.05, 0.89) [80] had higher odds of non-adherence to self-care practice.

Twelve primary studies [27,31,46–48,52,53,55–57,59,81] assessed the association between educational level and non-adherence to self-care practice. Three primary studies [45,52,53] found that male people with DM were more likely to non-adherence to self-care practice than female people with DM (AOR: 1.84, 95% CI: 1.04, 2.64). Three [48,53,61], four [27,31,45,58] and five [31,52,57,58,61] primary studies were reported age, residency, and income were associated with non-adherence to self-care practice respectively.

The pooled effect of male participants (Pooled Odds Ratio (POR): 1.84; 95%CI: 1.04–3.92; $I^2$ = 51.2%), and participants with low level of literacy (POR: 2.89; 95%CI: 1.52–4.25; $I^2$ = 0.0%) were significantly affect on non-adherence to self-care practice in people with diabetes. However, no significant association between age (POR: 1.49; 95% CI: -0.95–3.92, $I^2$ = 51.2%), residency (POR: 0.48; 95% CI: -0.26-.21, $I^2$ = 0.0%) and income (AOR: 0.85; 95% CI: 0.15, 1.54, $I^2$ = 69.5%), and non-adherence to self-care practice were observed (Table 4).

**Access to DM education and glucometer.** The pooled effect of five primary studies [29,50,54,57] on diabetes self-management education revealed the association with non-adherence to self-care practice. People with DM who did not access diabetes self-management education had higher odds of non-adherence to self-care practice (POR: 2.76; 95% CI: 1.7–3.55, $I^2$

= 0.0%). Four [29,54,57,62] primary studies were examined the association between access to private glucometer and non-adherence to self-care practice. After pooling four primary studies, participants who had no private glucometer at home had higher odds of non-adherence to self-care practice (POR: 2.71; 95% CI: 1.46–3.95, $I^2$ = 0.0%). Eight primary studies [29,48,51,54,56,57,62] reported a relationship between knowledge towards diabetes and non-adherence to self-care practice. However, the pooled result showed no significant relationship between knowledge towards diabetes and non-adherence to self-care practice (POR: 1.48; 95% CI: 0.79–2.17, $I^2$ = 77.3) (Table 4).

**Clinical factors.** Studies investigated the association between clinical variables and non-adherence to self-care practice. These clinical factors are co-morbidity (AOR: 1.68; 1.07–2.56) [51], medication burden (AOR: 2.87; 1.43–5.76) [55], uncontrolled plasma glucose level (AOR: 1.68; 1.02–2.79) [27], the level of medication adherence (AOR: 0.4; 95%CI: 0.30–0.80), and alcohol consumption (AOR: 4.6; 95%CI: 1.30–15.7) [47]. The pooled effect of four [45,46,48,52] primary studies found that people who lived a shorter time with DM were more than three times more likely to have non-adherence to self-care practice than people who lived a long time with DM (POR: 3.69; 95% CI: 1.86–5.52, $I^2$ = 0.0%).

The pooled result of four [27,31,52,53] observational studies found that people who had one or more DM complication were more likely to non-adherence to self-care practice than those who did not have a DM complication (AOR: 2.22; 95% CI: 1.48–2.95, I2 = 0.0%). The pooled effect of two primary studies found that people who did not satisfy with medical care were more likely non-adherence to self-care practice than people who had medical care satisfied (POR: 1.8; 95% CI: 1.15–2.44, $I^2$ = 0.0%). Treatment modality did not affect non-adherence to self-care practice across two different observational studies [51,53] (POR: 1.19; 95% CI: -0.19–2.57, $I^2$ = 90.4%) (Table 4).

**Psycho-social factors.** Eight observational studies [29,31,50,55–57,81,82] were investigated on the association between psychosocial variables and non-adherence to self-care practice. People who had moderate perceived barriers towards self-care practice were 70% more likely non-adherence to self-care practice than people who had less perceived barriers towards self-care practice (AOR: 0.3; 95% CI: 0.12–0.64) [57]. People with DM who did not perceive the severity of diabetes and its complications were twelve times more non-adhere to self-care practice than highly perceived ones (AOR: 12.3; 95% CI: 1.19–13.25) [57]. People with DM who had an unfavorable attitude toward perceived benefit (AOR: 5.94; 1.87–17.94) and perceived barrier (AOR: 0.47; 0.27–0.39) were most likely to non-adherence to self-care practice than favorable attitude toward the perceived benefit and perceived barrier [29] respectively.

Seven primary studies [50,55,56,81–84] investigated the relationship between social support and non-adherence to self-care practice. Pooled results showed that social support did not affect non-adherence to self-care practice (POR: 1.31; 95% CI: 0.72–1.89, $I^2$ = 86.5%). Two observational studies [62,82] examined the association between self-efficacy of self-care practice and non-adherence to self-care practice. After pooling the two studies' result, people with diabetes who had poor self-efficacy towards self-care practice were more likely to self-care non-adherence than those who had good self-efficacy (POR: 3.09; 95% CI: 1.70–4.48, $I^2$ = 0.0%) (Table 4).

## Discussion

To the best of our review, this is the first systemic review and meta-analysis study that conducted to show the pooled proportion of non-adherence and factors associated with non-adherence to self-care practice in Ethiopian adults with diabetes.

This study identified that many people with diabetes (49.91%) in Ethiopia fail to adhere to self-care practices as recommended or agreed between people with diabetes and healthcare providers. This is supported by a systemic review in sub-Saharan Africa (49%), which is of poor quality of life and a serious threat to the health of people, and the health system capacity [85].

This is because chronic care health systems in sub-Saharan Africa have significantly developed in the last decade, the capacity for managing diabetes remains in its infancy, including in Ethiopia [86]. In addition to this, the international community is certainly aware of sub-Saharan Africa problems caused by malnutrition and HIV and other infectious diseases, awareness of the increasing burden of chronic non-communicable diseases including diabetes is still low. This turns into a vital part of self-care behaviors, health beliefs, attitudes toward diabetes, and self-efficacy underutilized in developing countries, including Ethiopia [87,88].

This finding was higher than a study in Bangladesh (28.2%) (28.2%) [89] and much lower than the study reported in Australia (85.5%) [90], Oman (73%) [91], and Iran (65.15%) [92]. Health care facility access, prevent the incidence and prevalence of type 2 diabetes, prevention, and control of diabetes have proven successful at identifying high-risk people, particularly in rural and remote-access areas, and screen and increase awareness of people about diabetes on special days, and places. These could be the reasons for the difference. Another issue related to the healthcare system is a tremendous mismatch between supply and demand in Ethiopia. This is because of the unavailability of the system for the continuity of DM care. For instance, inadequate knowledge and skills of health workers, frequent stock-outs of DM supplies, and people with diabetes inability to pay for DM related medical service [93].

In Oman, diabetes care focuses on community resources, well-designed health system organization of diabetes health care, self-management support, delivery system design, decision support, and clinical information systems. The execution of these types of the system by healthcare authorities in Oman is a key element in the success of diabetes care [94]. Whereas in Iran, the healthcare system is the referral feedback loop from academic specialized centers back to the lowest levels of healthcare facility helped support the continuum of care for people with pre-diabetes and diabetes that is a primary level (diabetes unit) to secondary level (diabetes center) to tertiary level (specialized care) and vice-versa [95].

When to see the Australian diabetes care, the Australian National Diabetes Strategy 2016–2020 sets different goals to prevent and control diabetes. Some of these are preventing people from developing type 2 diabetes, promoting awareness and earlier detection of type 1 and type 2 diabetes, reducing the occurrence of diabetes-related complications and improving quality of life among people with diabetes, and strengthen prevention and care through research, evidence, and well-organized database [96]. Diabetes care in Bangladesh was seeking pharmaceutical therapy and dependence on health professional care rather than self-management. That is why diabetes significantly increases healthcare use and expenditure and is likely to impose a huge economic burden on the healthcare systems in Bangladesh [97,98].

Non-adherence to recognized standards of care is the principal cause of the development of complications of diabetes and associated people, societies, and economic costs [99]. Non-adherence to diabetes self-care could lead to short-term complications and uncontrolled diabetes [100] which results in avoidable suffering for people with diabetes and excess cost to the healthcare systems too [101,102]. Uncontrolled diabetes increases susceptibility to infection and worsens outcomes for some of the world's major infectious diseases including tuberculosis and Melioidosis [103].

This study indicated that socio-demographic related characters (age, sex, family history, and income), overall health beliefs, clinically related characters (treatment modality, complications, and satisfaction with medical care), self-efficacy, social support, and diabetes self-

management education about diabetes were factors consistently reported in the literature associated with diabetes self-care practice.

Sex is one of the potential factors for non-adherence to self-care practice. Males with diabetes were non-adhered to self-care practice more than women. This supported by a qualitative study in Pakistan [104], Mexico [105], and Indonesia [106]. This is due to women being more adaptable to diabetes conditions, disclosing their diabetes more readily, and being a lot more ready to integrate management into their daily lives. They also used socially interactive resources such as internet access and education classes, because men relied more on self-directed learning, more reluctant to tell friends and family about their diabetes due to the stigma attached to the disease, and were less observant of self-management practices in social settings [107].

In this study, the educational level is a potential reason. Those people with diabetes who have low levels of literacy had more likely to self-care non-adherence behaviors. A previous systemic review with a meta-analysis study done in China suggested that participants with a higher educational level were linked to better diabetes self-care practice [108]. Being higher education levels could lead to having better judgment and decision-making ability for adhering to self-care behaviors [92].

Diabetes Self-Management Education (DSME) associated with self-care practice. In line with earlier studies [85,92,109,110], people with diabetes who had no access to diabetes self-management education were associated with non-adherence to self-care practice. Not access to diabetes self-management education associated with inadequate diabetes self-care practice [111]. Diabetes self-management education plays an important role in self-care for all people with diabetes and is necessary to improve patient outcomes [112]. It has also facilitated the knowledge, skill, and ability necessary for diabetes self-care practices. This is having a positive impact on people with diabetes access, use, and outcomes [113].

National standards for DSME and support recommended that all people with diabetes should take part in diabetes self-management education to ease the knowledge, skills, and ability necessary for diabetes self-care and in diabetes self-management support to help with implementing and sustaining skills and behaviors needed for ongoing self-management [114]. Given the impact of DSME in developing countries, it has positive effects on knowledge, glycemic control, and behavioral outcomes on short-term follow-up. DSME on people with diabetes' self-care practice can reduce complications like arterial stiffness, improve quality of life, and improve their self-care behaviors [115,116].

We found that the proportion of non-adherence to self-care practice is more pronounced among people with diabetes who did not have glucometer access at their home. This estimation is precise as a confidence interval ranged from 1.70 (almost double effect on the odds of non-adherence to self-care practice) to 3.55 (almost four times the effect). This supported by a systemic review was done in Sub-Saharan Africa [85] and a qualitative study done in Ghana [117]. This may have a serious financial implication (the high cost of blood glucose meter and reagent strips) and supplies (no access to blood glucometer) [118].

Self-monitoring of blood glucose is an important practice for people with diabetes. Self-monitoring of blood glucose systems has the potential to play an important role in the management of diabetes and in the reduction of risks of serious clinical complications. Therefore, effective balanced costs of blood glucose meter and reagent strips are required to support and potentially cost-saving for people with diabetes. It is also promoted easy access to glucometer and reagents which are offered free of charge to people with diabetes to increase self-care adherence [118–120].

People with diabetes living for with shorter time with diabetes are almost quadrupled prone to non-adherence to self-care practice than people with diabetes living for a long

time. The experience of living with diabetes can take significant sounds on the well-being of people in terms of the success of survival skills with the disease living a long duration. Individuals with diabetes who have lived a long time with diabetes realize, master the basic skills and information, acquire in-depth and advanced diabetes knowledge throughout their lifetime, both formally through programs of continuing education and informally through experience and sharing of information with other people with diabetes. Individuals with positive coping strategies such as confrontation tend to more proactive in learning to manage their disease.

In contrast, newly diagnosed and/or short-live with diabetes is challenged to master the basic skills and information and take time to acquire in-depth and advanced diabetes knowledge. The new diagnosis of people with diabetes may have negative coping strategies such as avoidance or acceptance-resignation. They may not be willing to follow management recommendations. Therefore, diabetes educators should focus on helping newly diagnosed people with diabetes to develop positive coping strategies to adhere to self-care practices [108,121].

We found that participants who had one or more DM complications were more often non-adherence to self-care practice than those who did not have DM complications. This finding is not in line with the previous study done in China [108]. This difference is due to the difference between health service access (multi-discipline approach in China's health service use modern and traditional approach), different study populations (type 2), and different sociocultural characteristics of participants.

We found that participants who did not satisfied with their medical care was more likely non-adherence to self-care practice than their counterparts. In line with earlier studies, dissatisfaction with medical care restricted self-care activities in developing countries including Ethiopia [122]. This is due to the barrier of medical care satisfaction is a part of the provider's perspective (belief by providers that recommended self-care practice cannot cure/help clients, no confidence in their own ability to alert client behaviors and inadequate training in standard diabetes car), client factors (attitude, belief, knowledge about diabetes, frustration due to dynamic and chronic nature of diabetes, financial resources and co-morbidity) and clinical environment (health facility readiness, management systems for service and unequal distribution of health providers between different health facilities) [117,123–125].

Therefore, interventions to improve adherence to diabetes self-care should focus on helping people with diabetes develop a favorable attitude and how to overcome behavioral control barriers. In addition to this, improving access to standard diabetic care, basic technologies, and competence of health workers can improve the care process leading to better outcomes.

People with diabetes who had poor self-efficacy towards self-care practice were to have non-adherence to self-care practice than those who had good self-efficacy. This was supported by the previous study [108], there was a consistently strong association between poor self-efficacy level and non-adherence to self-care practice in people with diabetes. This implies that the highest resistance to diabetes self-care practice, the less confidence the people with diabetes are in their ability to adhere to self-care practice recommendations. Therefore, the clinical implication is that self-efficacy in people with diabetes the first step in the development of people's tailored adherence to self-care practice.

## Strength and limitations

This systematic review and meta-analysis have several strengths. It is the first study that pooled the results of several studies in the country giving stronger evidence of different factors that restrict self-care practice among people with diabetes. It was able to include a relatively large

number of people with diabetes (N = 7,134), which is much more than sample sizes of each study. Also tried to find many factors that hinder the recommended self-care practice in people with diabetes.

Despite its strengths, the study also has a few limitations. Even though most of the studies had good quality, all of the primary studies were cross-sectional, which is limited in this study. In addition, although extensive and diverse search strategies were used to find all possible available literature, some grey literature, such as conference proceedings, was still difficult to find. Furthermore, the study was not able to conduct a subgroup analysis based on the different domains of self-care practice which would have given important information to find a potential target to improve self-care practice.

## Implication

This study has many implications for clinical practice and future research. Firstly, the health care provider (especially physicians and nurses) can develop effective strategies to improve certain diabetes self-care practice behaviors that people with DM exhibit, such as non-adherence to self-care practice. Secondly, identifying and understanding factors that restrict diabetes self-care practice is the first step in developing evidence-based interventions to promote short and long-term health outcomes and quality of life. Future research should focus on developing and testing a conceptual model that can use to enhance diabetes self-care practice in a national context. Finally, to give a long-term reduction in diabetes-related co-morbidity and mortality, researches should assess ways to extend and sustain diabetes self-care practice among this population.

## Conclusions

This meta-analysis revealed that a high proportion of people with diabetes failed to adhere to self-care practices. Socio-demographic related characters (age, sex, family history, income level, and residency), overall health beliefs, clinically related characters (treatment modality, complications, satisfaction with medical care and knowledge towards diabetes disease), self-efficacy, social support, and diabetes self-management education was hinder factors consistently reported in the literature with diabetes self-care practice.

## Supporting information

**S1 File. Protocol and registration.**
(DOCX)

**S2 File. PRISMA check list.**
(DOCX)

**S3 File. Search strategy applied to PubMed database.**
(DOCX)

**S4 File. Risk of bias assessment.**
(DOCX)

**S5 File. Excluded article.**
(PDF)

**S6 File. Registration and protocol.**
(PDF)

## Author Contributions

**Conceptualization:** Teshager Weldegiorgis Abate.

**Formal analysis:** Teshager Weldegiorgis Abate.

**Methodology:** Teshager Weldegiorgis Abate.

**Resources:** Emiru Ayalew.

**Writing – review & editing:** Getenet Dessie, Yinager Workineh, Haileyesus Gedamu, Minyi-chil Birhanu, Emiru Ayalew, Mulat Tirfie, Aklilu Endalamaw.

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
