## [Decision Letter · Decision Letter 0]

24 Nov 2020

PONE-D-20-23361

Self-care Non-adherence and Associated Factors among Diabetes Adult Population in Ethiopian: Systemic Review with Meta-Analysis

PLOS ONE

Dear Dr. Abate,

Thank you for submitting your manuscript to PLOS ONE. After careful consideration, we feel that it has merit but does not fully meet PLOS ONE’s publication criteria as it currently stands. Therefore, we invite you to submit a revised version of the manuscript that addresses the points raised during the review process.

The ms. addresses an interesting issue. However, it requires a very extensive revision according to the reviewer's comments, which need to be addressed entirely and properly.

We look forward to receiving your revised manuscript.

Kind regards,

Paolo Magni

Academic Editor

PLOS ONE

Journal Requirements:

2. Please change your reference to "p=0.000" to "p<0.001" or as similarly appropriate, as p values cannot equal zero.

3. Please change your reference to "p-value=NS to the exact value

4. Please explain the rationale for the search dates.

5.Thank you for stating the following in the Funding Section of your manuscript:

[The administrators of Bahir Dar University indirectly supported this project and had no role in study design, data collection, data analysis, data interpretation, or writing the report.]

 [No specific funding for this work ]

7.Please amend either the title on the online submission form (via Edit Submission) or the title in the manuscript so that they are identical.

8. Please amend the manuscript submission data (via Edit Submission) to include author Emiru Ayalew.

9. Please amend either the abstract on the online submission form (via Edit Submission) or the abstract in the manuscript so that they are identical.

10. Please remove your figures from within your manuscript file, leaving only the individual TIFF/EPS image files, uploaded separately.  These will be automatically included in the reviewers’ PDF.

11. We note that this manuscript is a systematic review or meta-analysis; our author guidelines therefore require that you use PRISMA guidance to help improve reporting quality of this type of study. Please upload copies of the completed PRISMA checklist as Supporting Information with a file name “PRISMA checklist”.

Additional Editor Comments (if provided):

The ms. addresses an interesting issue. However, it requires a very extensive revision according to the reviewer's comments, which need to be addressed entirely and properly.

Reviewers' comments:

Reviewer's Responses to Questions

**Comments to the Author**

1. Is the manuscript technically sound, and do the data support the conclusions?

Reviewer #1: Partly

2. Has the statistical analysis been performed appropriately and rigorously? 

Reviewer #1: No

3. Have the authors made all data underlying the findings in their manuscript fully available?

Reviewer #1: Yes

4. Is the manuscript presented in an intelligible fashion and written in standard English?

Reviewer #1: Yes

5. Review Comments to the Author

Reviewer #1: Dear Editor,

I carefully read the article by Abate et al., which is interesting a quite well done.

My remarks are the following:

- In the abstract, among the results authors should include how many articles were included in each analysis.

- The manuscript is not balanced now. In particular, the background paragraph should be shortened.

- Authors state that they used a random effect model when Higgin's index was >89.8%. Based on my experience and also considering what authors wrote in their manuscript, this cut-off should be replaced with 50%. Then, authors should repeat the analysis by using a fixed-effect model when I^2 is < or = 50% and a random-effect model when I^2 is >50%.

- Authors should include as supplementary material a list of articles which were excluded from the analysis.

- Table 1 is not very accurate. Authors should enlist the included studies in alphabetical order or based on the publication date of the article.

- To enrich table 1, authors might include also the prevalence of man / woman in each single study.

- In the manuscript, authors should consider to refer to doi: 10.1016/j.numecd.2020.03.005.

6. PLOS authors have the option to publish the peer review history of their article (what does this mean?). If published, this will include your full peer review and any attached files.

Reviewer #1: No

---

## [Author Response · Author response to Decision Letter 0]

30 Nov 2020

We thank the reviewer and Academic Editor for kind words. We have revised our manuscript

based on the comments and provided response accordingly.

---

## [Decision Letter · Decision Letter 1]

8 Dec 2020

PONE-D-20-23361R1

Non-adherence to self-care and associated factors among diabetes adult population in Ethiopian: a systemic review with meta-analysis. 

PLOS ONE

Dear Dr. Abate,

Thank you for submitting your manuscript to PLOS ONE. After careful consideration, we feel that it has merit but does not fully meet PLOS ONE’s publication criteria as it currently stands. Therefore, we invite you to submit a revised version of the manuscript that addresses all the points raised during the review process.

We look forward to receiving your revised manuscript.

Kind regards,

Paolo Magni

Academic Editor

PLOS ONE

Additional Editor Comments (if provided):

The paper, although improved, still has major flaws that need to be fully addressed.

Reviewers' comments:

Reviewer's Responses to Questions

**Comments to the Author**

1. If the authors have adequately addressed your comments raised in a previous round of review and you feel that this manuscript is now acceptable for publication, you may indicate that here to bypass the “Comments to the Author” section, enter your conflict of interest statement in the “Confidential to Editor” section, and submit your "Accept" recommendation.

Reviewer #1: (No Response)

2. Is the manuscript technically sound, and do the data support the conclusions?

Reviewer #1: Partly

3. Has the statistical analysis been performed appropriately and rigorously? 

Reviewer #1: No

4. Have the authors made all data underlying the findings in their manuscript fully available?

Reviewer #1: Yes

5. Is the manuscript presented in an intelligible fashion and written in standard English?

Reviewer #1: Yes

6. Review Comments to the Author

Reviewer #1: Dear Editor,

even though the manuscript is significantly improved in comparison with the original version, I still have some concerns about the method of data synthesis used. As a matter of fact, authors should have used a random-effect model when heterogeneity was high and a fixed-effect model when heterogeneity was low, also in the subgroup analyses. If this approach was adopted, it was not clearly declared in the methods.

Then, I suggested the authors to discuss their results in the context of the findings published by Cicero et al. (PMID: 32249143), regarding the awareness of diabetes in a well characterized European population. In the rebuttal letter, authors said they could not find the article using the doi (?)... I do not dare to think how they could accurately have done the bibliographic search for their meta-analysis!

Finally, the current systematic review and meta-analysis has some deviations from its protocol (as it was registered in PROSPERO). This is a flaw that needs to be addressed.

7. PLOS authors have the option to publish the peer review history of their article (what does this mean?). If published, this will include your full peer review and any attached files.

Reviewer #1: No

---

## [Author Response · Author response to Decision Letter 1]

17 Dec 2020

Response to Reviewers

Response to editor and reviewer

we thank you and the reviewers for a thorough reading and constructive criticism of our manuscript and for the opportunity to revise and resubmit. We are pleased to submit the improved research article, including a proposed comment, “Non-adherence to self-care and associated factors among diabetes adult population in Ethiopian: a systemic review with meta-analysis”

RESPONSE TO REVIEWER 1

REVIEWER COMMENT: Has the statistical analysis been performed appropriately and rigorously? No

RESPONSE: we try to explain the statistical analysis appropriately and rigorously on page 8-9, lines 180-205 (including heterogeneity test, publication bias assessment and meta-regression, and sensitivity analysis)

REVIEWER COMMENT: even though the manuscript is significantly improved in comparison with the original version, I still have some concerns about the method of data synthesis used. As a matter of fact, authors should have used a random-effect model when heterogeneity was high and a fixed-effect model when heterogeneity was low, also in the subgroup analyses. If this approach was adopted, it was not clearly declared in the methods.

RESPONSE: this is an important comment. We incorporate the means of sub-group analysis the method section on page 9, lines 192-194.

REVIEWER COMMENT: I suggested the authors to discuss their results in the context of the findings published by Cicero et al. (PMID: 32249143), regarding the awareness of diabetes in a well characterized European population. In the rebuttal letter, authors said they could not find the article using the doi (?)... I do not dare to think how they could accurately have done the bibliographic search for their meta-analysis!

RESPONSE: we are discuss our results in the context of the findings published by Cicero et al in the discussion part.

REVIEWER COMMENT: finally, the current systematic review and meta-analysis has some deviations from its protocol (as it was registered in PROSPERO). This is a flaw that needs to be addressed.

RESPONSE: 

That was an editorial problem, we edited the protocol according to the current manuscript.

---

## [Decision Letter · Decision Letter 2]

11 Jan 2021

Non-adherence to self-care and associated factors among diabetes adult population in Ethiopian: a systemic review with meta-analysis. 

PONE-D-20-23361R2

Dear Dr. Abate,

We’re pleased to inform you that your manuscript has been judged scientifically suitable for publication and will be formally accepted for publication once it meets all outstanding technical requirements.

Kind regards,

Paolo Magni

Academic Editor

PLOS ONE

Reviewers' comments:

Reviewer's Responses to Questions

**Comments to the Author**

1. If the authors have adequately addressed your comments raised in a previous round of review and you feel that this manuscript is now acceptable for publication, you may indicate that here to bypass the “Comments to the Author” section, enter your conflict of interest statement in the “Confidential to Editor” section, and submit your "Accept" recommendation.

Reviewer #1: All comments have been addressed

2. Is the manuscript technically sound, and do the data support the conclusions?

Reviewer #1: Yes

3. Has the statistical analysis been performed appropriately and rigorously? 

Reviewer #1: Yes

4. Have the authors made all data underlying the findings in their manuscript fully available?

Reviewer #1: Yes

5. Is the manuscript presented in an intelligible fashion and written in standard English?

Reviewer #1: Yes

6. Review Comments to the Author

Reviewer #1: Dear Editor,

the last version of the manuscript is significantly improved in comparison with the original one. I recommend its acceptation in the Journal.

7. PLOS authors have the option to publish the peer review history of their article (what does this mean?). If published, this will include your full peer review and any attached files.

Reviewer #1: No

---

## [Editor Report · Acceptance letter]

15 Jan 2021

PONE-D-20-23361R2 

Non-adherence to self-care and associated factors among diabetes adult population in Ethiopian: a systemic review with meta-analysis. 

Dear Dr. Abate:

I'm pleased to inform you that your manuscript has been deemed suitable for publication in PLOS ONE. Congratulations! Your manuscript is now with our production department. 

Kind regards, 

on behalf of

Prof. Paolo Magni 

Academic Editor

PLOS ONE